# AN INVESTIGATION ON HARDWARE-AWARE VISION TRANSFORMER SCALING

## ABSTRACT

Vision Transformer (ViT) has demonstrated promising performance in various computer vision tasks, and recently attracted a lot of research attention. Many recent works have focused on proposing new architectures to improve ViT and deploying it into real-world applications. However, little effort has been made to analyze and understand ViT's architecture design space and its implication of hardware-cost on different devices. In this work, by simply scaling ViT's depth, width, input size, and other basic configurations, we show that a scaled vanilla ViT model without bells and whistles can achieve comparable or superior accuracy-efficiency trade-off than most of the latest ViT variants. Specifically, compared to DeiT-Tiny, our scaled model achieves a $\uparrow 1.9\%$ higher ImageNet top-1 accuracy under the same FLOPs and a $\uparrow 3.7\%$ better ImageNet top-1 accuracy under the same latency on an NVIDIA Edge GPU TX2. Motivated by this, we further investigate the extracted scaling strategies from the following two aspects: (1) "*can these scaling strategies be transferred across different real hardware devices*?"; and (2) "*can these scaling strategies be transferred to different ViT variants and tasks*?". For (1), our exploration, based on various devices with different resource budgets, indicates that the transferability effectiveness depends on the underlying device together with its corresponding deployment tool; for (2), we validate the effective transferability of the aforementioned scaling strategies obtained from a vanilla ViT model on top of an image classification task to the PiT model, a strong ViT variant targeting efficiency, as well as object detection and video classification tasks. In particular, when transferred to PiT, our scaling strategies lead to a boosted ImageNet top-1 accuracy of from 74.6% to 76.7% ($\uparrow 2.1\%$) under the same 0.7G FLOPs; and when transferred to the COCO object detection task, the average precision is boosted by $\uparrow 0.7\%$ under a similar throughput on a V100 GPU.

## 1 INTRODUCTION

Transformer (Vaswani et al., 2017), which was initially proposed for natural language processing (NLP) and is a type of deep neural networks (DNNs) mainly based on the self-attention mechanism, has achieved significant breakthroughs in NLP tasks. Thanks to its strong representation capabilities, many works have developed ways to apply Transformer to computer vision (CV) tasks, such as image classification (Dosovitskiy et al., 2021), object detection (Carion et al., 2020), semantic segmentation (Wu et al., 2020), and video classification (Bertasius et al., 2021). Among them, Vision Transformer (ViT) (Dosovitskiy et al., 2021) stands out and demonstrates that a pure Transformer applied directly to sequences of image patches can perform very well on image classification tasks, e.g., achieving a comparable ImageNet (Deng et al., 2009) top-1 accuracy as ResNet (He et al., 2016). Motivated by ViT's promising performance, a fast growing number of works follow it to explore pure Transformer architectures in order to push forward its accuracy-efficiency trade-off and deployment into real-world applications (Touvron et al., 2020; Heo et al., 2021; Graham et al., 2021; Liu et al., 2021; Wu et al., 2021), achieving an even better performance than EfficientNetV1 (Tan & Le, 2019), a widely used efficient convolutional neural network (CNN).

The success of recent ViT works suggests that the model architecture is critical to ViT's achievable performance. Therefore, in this work we explore ViT architectures from a new perspective, aiming to analyze and understand ViT's architecture design space and real hardware-cost across different

devices. Despite the recent excitement towards ViT models and the success of model scaling for CNNs, little effort has been made into exploring ViT's model scaling strategies or hardware-cost.

Note that directly applying the scaling strategies for CNNs (Tan & Le, 2019; Sun et al., 2017) or Transformer on NLP tasks (Kaplan et al., 2020; Henighan et al., 2020) will lead to sub-optimality, as discussed in Section 3.2. Furthermore, scaling strategies targeting one device/task might not be transferable to another device/task. Interestingly, we find that simply scaled ViT models can achieve comparable or even better accuracy-efficiency trade-off than dedicatedly designed ViT variants, as shown in Figure 1. Motivated by this, we further explore the transferability of our scaling strategies (1) across different real hardware devices and (2) to different ViT variants and tasks. In particular, we make the following contributions:

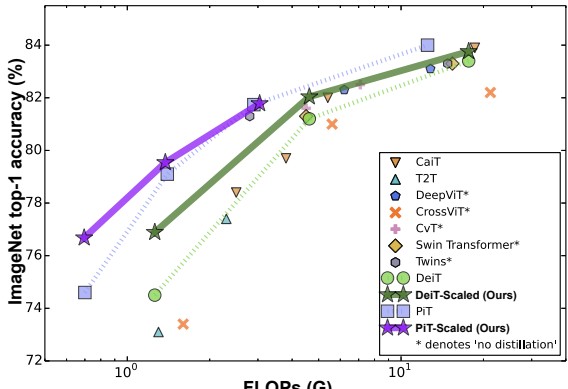

Figure 1: Our scaled ViT models achieve comparable or better accuracy-efficiency trade-off as compared to some recent dedicatedly designed ViT variants.

- We are the first to show that simply scaled vanilla ViT models can achieve comparable or even better accuracy-efficiency trade-off as compared to dedicatedly designed ViT variants (Touvron et al., 2021; Yuan et al., 2021; Zhou et al., 2021; Chen et al., 2021; Wu et al., 2021; Liu et al., 2021; Chu et al., 2021; Touvron et al., 2020; Heo et al., 2021), as illustrated in Figure 1. Specifically, as compared to DeiT-Tiny, our scaled model achieves a ↑ 1.9% higher ImageNet top-1 accuracy under the same FLOPs and a ↑ 3.7% better ImageNet top-1 accuracy under the same latency on an NVIDIA Edge GPU TX2.

- We study the transferability of the scaled ViT models across different devices and show that the transferability effectiveness depends on the underlying devices and deployment tools. For example, scaling strategies targeting FLOPs or the throughput on V100 GPU (NVIDIA LLC.) can be transferred to the Pixel3 (Google LLC., a) device with little or even no performance loss, but those targeting the latency on TX2 (NVIDIA Inc., c) may not be transferred to other devices due to the obvious performance loss. Additionally, we provide ViT models' cost breakdown and rank correlation between their hardware-cost on different devices for better understanding it.

- We show that our scaling strategies can also be effectively transferred to different ViT variants and recognition tasks to further boost the achieved accuracy-efficiency trade-off, e.g., achieving a ↑ 2.1% higher accuracy under a similar FLOPs when being transferred to the PiT model and ↑ 0.7% higher average precision under a similar inference throughput when being transferred to an object detection task.

## 2 RELATED WORKS

**Vision Transformers.** Transformer was first proposed for machine translation (Vaswani et al., 2017). Motivated by its state-of-the-art (SOTA) performance in NLP tasks, there has been a growing interest in applying the Transformer/self-attention mechanism to CV tasks, e.g., by proposing novel attention mechanisms for CNNs (Hu et al., 2018; Li et al., 2019; Zhang et al., 2020), fusing Transformer and CNN designs within the same model (Bello et al., 2019; Carion et al., 2020; Wu et al., 2020), or designing pure Transformer models (Dosovitskiy et al., 2021; Chen et al., 2020). Among them, ViT (Dosovitskiy et al., 2021) has achieved SOTA performance by directly applying the Transformer architecture for NLP tasks to the input raw image patches of vision tasks. Nevertheless, ViT's powerful performance largely depends on its pre-training on JFT-300M (Sun et al., 2017) (a giant private labelled dataset). As such, DeiT (Touvron et al., 2020) further develops an improved training recipe (i.e., the setting of optimization hyper-parameters), including a distillation setup and stronger data augmentation and regularization, to achieve comparable performance while removing the necessity of the costly pre-training. In order to build more efficient ViT models, (Chen et al., 2021) leverages multiple branches to extract and fuse features at different scales; (Heo et al., 2021; Graham et al.,

2021; Liu et al., 2021; Fan et al., 2021; Wang et al., 2021) apply a pyramid-like architecture commonly used in CNNs to ViT; and (Graham et al., 2021; Liu et al., 2021; Wu et al., 2021) propose more efficient attention mechanisms or feature projection blocks.

**Model scaling.** Prior works have explored scaling CNNs/NLP-Transformer (i.e., Transformer in NLP tasks) to boost its accuracy or lower its computational resource requirements, e.g., ResNet can be scaled along its depth dimension (He et al., 2016) and MobileNets can be scaled along its width (i.e., the number of channels) and input resolution dimensions (Sandler et al., 2018; Howard et al., 2019). Notably, EfficientNet further points out that it is critical to scale CNNs in a compound manner (i.e., simultaneously scaling the model width, depth, and input resolution) and does so to achieve SOTA accuracy-efficiency trade-off (Tan & Le, 2019). Nevertheless, as (Bello et al., 2021) demonstrates, the scaling strategies obtained from a specific model (e.g., EfficientNet-B0) can result in a sub-optimal accuracy-efficiency trade-off for another model; motivated by this observation, they develop a more general scaling strategies extracted from grid search experiments based on the chosen training recipe rather than a specific model, achieving an improved trade-off. In addition to scaling the model architecture, (Kaplan et al., 2020; Henighan et al., 2020) show that scaling up the dataset size and the number of computations used for training can also help to achieve a smaller cross-entropy loss for Transformer in NLP tasks. Recently, Zhou et al. (2021) demonstrates that the accuracy of ViT will decrease when it is scaled up along only the depth dimension (i.e., number of layers), and proposes Re-attention to resolve it.

Nevertheless, none of the prior works has targeted scaling strategies for ViT with multiple scaling factors or study its real-hardware efficiency across different platforms featuring diverse computational and storage capabilities. Additionally, it is not clear whether their insights on scaling CNNs can be directly applied to ViT because of their different scaling factor definitions, e.g., while the number of channels represents the width in CNNs, the number of heads and embedding dimensions can both represent the width in ViT. As such, scaling strategies dedicated to ViT are highly desirable and our scaling strategies can provide unique insights to inspire more innovations towards efficient ViT models. Although there is some model scaling strategy explorations in (Dosovitskiy et al., 2021; Zhai et al., 2021), our work distinguishes with them in providing more discoveries and insights. Specially, we focus more on the accuracy vs. efficiency trade-off when scaling ViTs instead of merely the accuracy; and (2) provide additional analysis on the transferability of the extracted scaling strategies across different devices, ViT variants, and tasks.

## 3 SCALING VIT: HOW AND WHY DO WE SCALE VIT?

In this section, we first analyze the scaling factors of ViT, then study the effectiveness of prior scaling strategies, which are dedicated to CNNs or Transformers, on ViT, and finally present our iterative greedy search approach to scale ViT.

### 3.1 SCALING FACTORS IN VIT

As analyzed in (Kaplan et al., 2020), the scaling factors in Transformers include the number of layers ($d$), the number of heads ($h$), the embedding dimension for each head ($e$), and the linear projection ratio ($r$). ViT, which directly adopts the Transformer architecture for NLP tasks and splits the raw images into patches to serve as the Transformer input, adds additional scaling factors, including image resolution ($I$) and patch size ($p$). Figure 2 illustrates and summarizes our considered scaling factors for ViT.

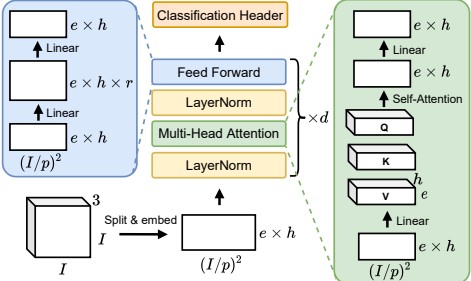

Figure 2: Illustrating the effect of scaling factors on a ViT architecture (class/distillation token is omitted for better visual clarity).

### 3.2 PREVIOUS SCALING STRATEGIES FAIL ON VIT

**CNN and ViT scaling factors do not match.** Scaling strategies dedicated to CNNs (Tan & Le, 2019; Sun et al., 2017; Feichtenhofer, 2020) mostly come with CNN-specific scaling factor definitions (e.g., the number of channels in convolution layers represents the model width), which cannot be directly transferred to ViT. For example, doubling ($2\times$) the width in CNNs can be achieved via various combinations of the number of heads ($h$) and embedding dimension for each head ($e$) in ViT.

Furthermore, there are extra scaling factors for ViT, e.g., the linear projection ratio ($r$) and the patch size ($p$), which do not exist in the scaling factors for CNNs but are important for ViT as shown in Appendix D, thus directly transferring the scaling strategies from CNNs to ViTs can lead to ambiguity and sub-optimal performance.

**Transformer scaling strategies for NLP is sub-optimal on ViT.** (Kaplan et al., 2020) noted that for NLP, model performance (i.e., accuracy or training loss) depends "strongly on the model scale (i.e., the number of parameters), but weakly on the model shape". However, when scaling ViT along the factors summarized in Figure 2, our observations suggest that this is not true for ViT. As shown in Figure 3, when performing an extensive search on top of DeiT-Small (Touvron et al., 2020) following (Sun et al., 2017), we observe that a model's shape has a great impact on the performance. Specifically, if we change the aspect ratio, i.e., the ratio between the embedding dimension ($e \times h$) and the number of layers ($d$), while keeping the model parameters to be the same, the accuracy drifts as much as 18.61%. This set of experiments motivates exploring scaling strategies dedicated to ViT.

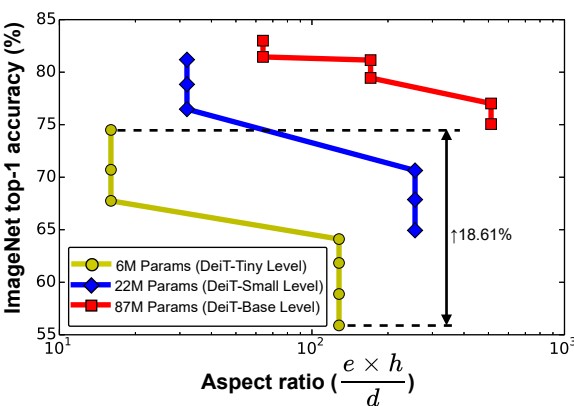

Figure 3: The accuracy of ViT is sensitive to the aspect ratio. Note that the vertically aligned points are models with the same scaling factors except image resolution ($I$).

### 3.3 OUR SCALING METHOD BASED ON AN ITERATIVE GREEDY SEARCH

Starting from a relatively small model defined in Table 1, we adopt a simple iterative greedy search to perform the ViT scaling step by step, similar to the previous algorithms for exploring CNN design spaces and feature selections (Feichtenhofer, 2020; John et al.; Jain & Zongker, 1997; Guyon & Elisseeff, 2003). Specifically, based on the starting point model or the optimal one from the previous scaling step, we scale up the model along each standalone scaling factor introduced in Section 3.1 to match the target hardware-cost (e.g., FLOPs, or latency on a specific hardware device), and select the one with the best accuracy-efficiency trade-off to be the starting point model of the next step. As analyzed in (Bello et al., 2021), unlike scaling

Table 1: The starting point model for our scaling method.

| | |
|---|---|
| Num. of layers ($d$) | 6 |
| Num. of heads ($h$) | 2 |
| Embedding dim. per head ($e$) | 64 |
| Linear projection ratio ($r$) | 4 |
| Image resolution ($I$) | 160 |
| Patch size ($p$) | 16 |
| FLOPs (G) | 0.15 |
| Throughput on V100 (FPS) | 20086 |
| Latency on Pixel3 (ms) | 30.05 |
| Latency on TX2 (ms) | 4.42 |

strategies extracted from a specific model, scaling based on such an iterative greedy search can avoid the unscalability of the resulting scaling strategies on a specific model. Our experiments in Section 4.1 also verify that such a scaling method is simple yet effective for scaling ViT models, and only requires training a few models during each search step.

## 4 EXPERIMENT RESULTS

In this section, we first present experiments for evaluating the scaled vanilla ViT models resulting from the iterative greedy search described in Section 3.3, in terms of accuracy-FLOPs trade-offs on ImageNet (Deng et al., 2009). From this set of experiments, we then extract a set of scaling strategies dedicated to ViT. After that, we further conduct experiments to study the transferability of our extracted scaling strategies (1) across different devices and (2) to different ViT variants and tasks.

### 4.1 SCALING VIT TOWARDS BETTER ACCURACY-FLOPS TRADE-OFFS

Following the scaling approach described in Section 3.3, we set $2 \times$ FLOPs of the initial or selected model from the previous step as the target hardware-cost in each step when individually scaling each factor, as summarized in Figure 2. All networks are trained for 300 epochs on ImageNet (Deng et al., 2009) using the same training recipe with the one in DeiT (Touvron et al., 2020), more details are included in Appendix E. We summarize our observations as follow:

Table 2: Our scaled ViT models outperform DeiT on ImageNet under the same FLOPs constrains.

| Model | FLOPs (G) | Top-1 accuracy (%) | $d$ | $h$ | $e$ | $r$ | $I$ | $p$ |
|---|---|---|---|---|---|---|---|---|
| DeiT-Tiny | 1.26 | 74.5 | 12 | 3 | 64 | 4 | 224 | 16 |
| **DeiT-Scaled-Tiny** | 1.22 | **76.4 (↑1.9)** | **14** | **4** | 64 | 4 | **160** | 16 |
| DeiT-Small | 4.62 | 81.2 | 12 | 6 | 64 | 4 | 224 | 16 |
| **DeiT-Scaled-Small** | 4.79 | **81.6 (↑0.4)** | **20** | **4** | 64 | 4 | **256** | 16 |
| DeiT-Base | 17.66 | 83.4 | 12 | 12 | 64 | 4 | 224 | 16 |
| **DeiT-Scaled-Base** | 16.82 | **83.8 (↑0.4)** | **20** | **6** | 64 | 4 | **320** | 16 |

**Scaled ViT models outperform SOTA DeiT models.** As shown in Table 2, our scaled ViT models (e.g., DeiT-Scaled-Tiny/Small/Base) achieve a ↑0.4% ∼ ↑1.9% higher top-1 accuracy on ImageNet under the same FLOPs constraints. Specifically, our DeiT-Scaled-Tiny model chooses to use a *smaller image resolution* (i.e., 160×160 vs. 224×224) and *more layers and a higher number of heads* as compared to the SOTA DeiT-Tiny model, and thus achieves a ↑1.9% higher accuracy at the same cost in terms of FLOPs, while our DeiT-Scaled-Small/Base models choose to use a *larger image resolution* (i.e., 320/256×320/256 vs. 224×224) and *more layers*, together with *a lower number of heads* as compared to the SOTA DeiT-Small/Base model, helping them to achieve a ↑0.4% higher accuracy under similar FLOPs. This set of experiments shows that our simple search method can (1) effectively locate ViT models with better accuracy-FLOPs trade-offs and (2) automatically adapt different scaling factors towards the optimal accuracy-FLOPs trade-offs, e.g., different model shapes and structures at different scales of FLOPs.

**Random permutation further boosts the performance.** Inspired by the coarse-to-fine architecture selection scheme adopted in (Yu et al., 2020), we further randomly permute the scaling factors (i.e., $d$, $h$, $e$, $r$, $I$, and $p$) of each scaled model in Table 2. After the permutation, we select **24** architectures under the same target hardware-cost with the scaled model by iterative greedy search for each scaled model. Figure 4 demonstrates that (1) such a random permutation can slightly push forward the frontier of accuracy-FLOPs trade-off (e.g., a ↑0.4% higher accuracy under similar FLOPs on top of the scaled models resulting from the adopted simple scaling method); and (2) our adopted iterative greedy search alone is sufficiently effective while requiring a lower exploration cost (e.g., 6 vs. 30 (6+24) models to be trained for each step as compared to such a search method together with the aforementioned permutation).

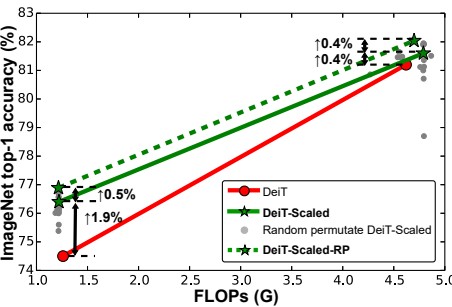

Figure 4: Random permutation on top of the DeiT-Scaled, where those on the Pareto frontier are marked as DeiT-Scaled-RP.

Table 3: Scaled ViT models after training for 1000 epochs.

| Model | FLOPs (G) | Top-1 accuracy (%) |
|---|---|---|
| DeiT-Tiny | 1.26 | 74.5 |
| **DeiT-Scaled-Tiny** | 1.22 | **76.4 (↑1.9)** |
| DeiT-Tiny / 1000 epochs | 1.26 | 76.6 |
| **DeiT-Scaled-Tiny / 1000 epochs** | 1.22 | **78.3 (↑1.7)** |
| DeiT-Small | 4.62 | 81.2 |
| **DeiT-Scaled-Small** | 4.79 | **81.6 (↑0.4)** |
| DeiT-Small / 1000 epochs | 4.62 | 82.6 |
| **DeiT-Scaled-Small / 1000 epochs** | 4.79 | **82.9 (↑0.3)** |

**Scaled ViT also benefits from a longer training time.** As pointed out by (Touvron et al., 2020), training ViT models for more epochs (e.g., 1000 epochs) can further improve the achieved accuracy. To verify whether the scaled ViT models can benefit from more training epochs, we train the models in Table 2 for 1000 epochs following the training recipe in (Touvron et al., 2020). As shown in Table 3, longer training epochs also help our scaled models (e.g., DeiT-Scaled-Tiny/Small) to achieve a higher accuracy, and thus, the advantage of our scaled models over DeiT is consistent under both the 300-epochs training recipe and 1000-epochs training recipe, e.g., a ↑1.9% higher accuracy over DeiT-Tiny *with 300 epochs* vs. a ↑1.7% higher accuracy over DeiT-Tiny *with 1000 epochs*.

**Drawn insights from scaling ViT.** Based on the observations from the above experiments, especially the scaling strategies illustrated in Figure 5, we draw the following scaling insights dedicated to ViT:

(1) When targeting relatively small models (i.e., with smaller FLOPs than DeiT-Scaled-Small), the optimal models tend to select "scaling $h$ (i.e., the number of heads)" or "scaling $d$ (i.e., the number of layers)" and a "smaller $I$ (i.e., the input image resolution)" (e.g., $160 \times 160$ instead of the commonly used $224 \times 224$).

(2) When targeting relatively large models (i.e., with larger FLOPs than DeiT-Scaled-Small), the optimal models mainly select to "scaling $I$ (i.e., the input image resolution)", while "slowing down scaling $h$ (i.e., number of heads)" as compared to the case when targeting relatively small models.

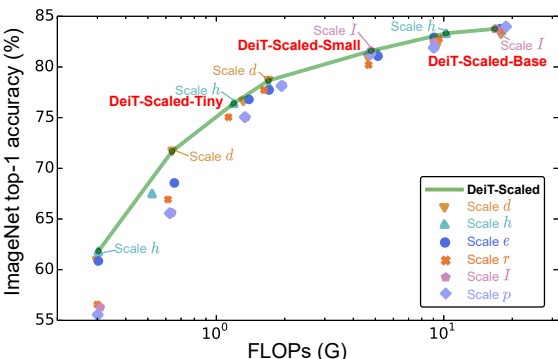

Figure 5: Resulting models from our iterative greedy search, where models achieving the best accuracy-FLOPs trade-offs are marked as DeiT-Scaled-Tiny/Small/Base. The architecture configurations (i.e., sets of $d$, $h$, $e$, $r$, $I$, and $p$) leading to these best models are extracted as our scaling strategies dedicated to ViT.

### 4.2 TRANSFERABILITY OF THE EXTRACTED SCALING STRATEGIES ACROSS DIFFERENT DEVICES

To evaluate the transferability of the extracted scaling strategies across different real hardware devices, we consider 3 hardware devices which target different applications as summarized in Table 4. More details about the setup of these devices are provided in Appendix B.

#### 4.2.1 TRANSFERABILITY AMONG DIFFERENT DEVICES

To obtain the hardware-dedicated scaling strategies leading to the best accuracy-efficiency trade-off on each device, we follow the scaling search method described in Section 4.1, but replace the target hardware-cost with (1) $0.5\times$ throughput measured on an NVIDIA V100 GPU (i.e., V100) (NVIDIA LLC.) (2) $2\times$ latency measured on an NVIDIA Edge GPU TX2 (i.e., TX2) (NVIDIA Inc., c), and (3) $2\times$ latency measured on a Google Pixel3 device (i.e., Pixel3) (Google LLC., a), to simulate the model scaling for (1) cloud services with strong GPUs, (2) edge computing with weak GPUs, and (3) mobile deployment without GPUs, respectively. We then compare the scaled models that achieve the best accuracy-efficiency trade-off on each device, as shown in Figure 6, aiming to answer "*can our scaling strategies be transferred across different real hardware devices*?". This set of comparisons provides some interesting observations:

(1) **The simple scaling approach is effective on different hardware devices.** From the comparison between the scaled models with FLOPs, throughput on V100, latency on TX2, and latency on Pixel3 as the hardware-cost during scaling (i.e., FLOPs (⋆), V100 (▲), TX2 (⬟), and Pixel3 (◆) Scaling in Figure 6) and the SOTA DeiT model (i.e., Baseline (●) in Figure 6), as shown in Figure 6) (a), (b), (c), and (d), respectively, we can see that all the device-dedicated scaled models resulting from the iterative greedy search method described in Section 3.3 achieve a better accuracy-efficiency trade-off than the baseline DeiT, indicating the necessity of device-dedicated scaling. Specifically, the scaled models targeting the TX2 device (⬟) can achieve a ↑ 3.7% higher accuracy under a similar latency on TX2, as compared to the DeiT-Tiny model. This set of experiments verifies that the adopted scaling approach is simple yet effective across different devices or targeting hardware metrics.

(2) **The transferability of our scaling strategies across different devices depends on the underlying device.** From Figure 6, we can observe that (i) the scaled models directly targeting a device indeed always lead to the best accuracy-efficiency trade-off on the device, indicating that our scaling search method can adapt to different devices; and (ii) the performance of the device-dedicated

Table 4: Important details about the 3 hardware devices in the transferability exploration experiments.

| Device | Deployment tool | Hardware-cost measurement tool | Target application |
|---|---|---|---|
| NVIDIA V100 | PyTorch | PyTorch profiler | Cloud services w/ strong GPUs |
| NVIDIA Edge GPU TX2 | TensorRT | TensorRT command-line wrapper | Edge computing w/ weak GPUs |
| Google Pixel3 | Tflite | Tflite benchmark tools | Mobile deployment w/o GPUs |

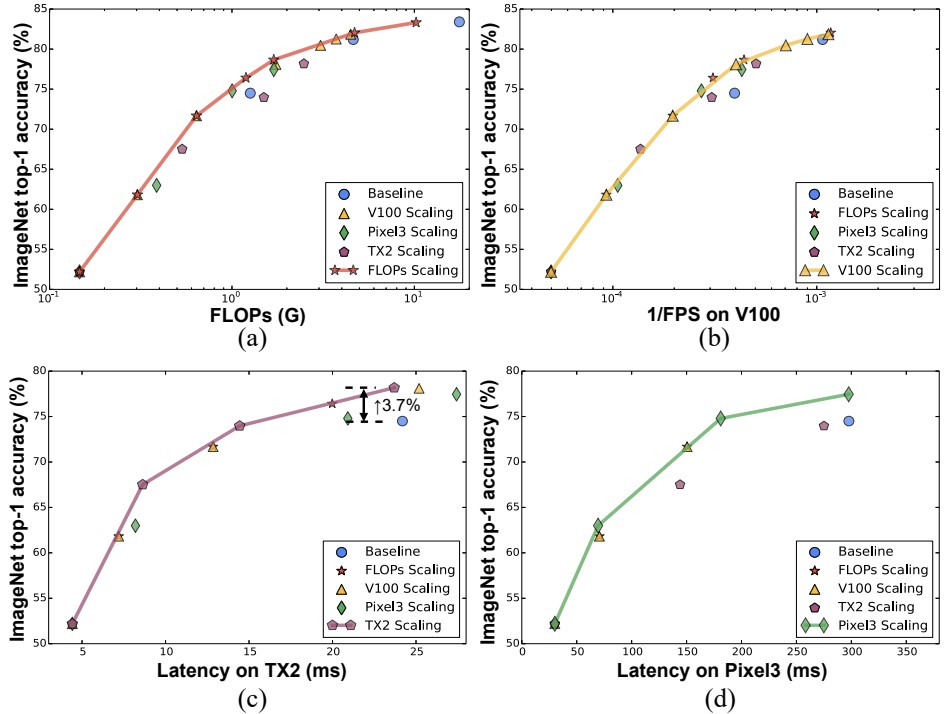

Figure 6: Comparing the optimal models resulting from scaling for different hardware devices. *FLOPs/V100/TX2/Pixel3 Scaling* represents the scaling strategies obtained on FLOPs/V100/TX2/Pixel3, and DeiT models are marked as the comparison Baseline.

scaled models when executed on other devices varies among different devices together with their corresponding deployment tools. For example, when executed on the Pixel3 device (see Figure 6 (d)), as expected, the scaled models targeting the Pixel3 device (denoted as ◆) are always on the Pareto frontier (i.e., the best accuracy-efficiency trade-off); interestingly, the scaled models targeting FLOPs (⋆) and the V100 device (▲) are also close to or even on the Pareto frontier; however, the scaled models targeting the TX2 device (⬠) are obviously far from the Pareto frontier when executed on the Pixel3 device.

As shown in Table 5, the scaled model targeting the TX2 device (⬠) suffers from a ↓ 0.82% lower accuracy at an even ↑ 51.90% higher latency when executed on the Pixel3 device, as compared to the scaled models directly targeting the Pixel3 device (◆), and vice versa for the performance of the scaled models targeting the Pixel3 device (◆) when executed on the TX2 device, i.e., a ↑ 15.74% higher latency and a ↓ 0.72% lower accuracy.

This set of experiments indicates that the scaling strategies obtained when targeting FLOPs (⋆) and the V100 device (▲) can be transferred to the Pixel3 device with little or even no performance loss, but those obtained for the TX2 device (⬠) leads to a degraded performance when being transferred.

### 4.2.2 ANALYSIS ON THE TRANSFERABILITY EFFECTIVENESS

To better understand why the transferability effectiveness depends on the underlying devices, we analyze the performance of ViT models executed on different hardware devices from the following

Table 5: Scaled models targeting Pixel3 are sub-optimal when executed on TX2, and vice versa.

| Model | Top-1 accuracy (%) | Latency on TX2 (ms) | Latency on Pixel3 (ms) | $d$ | $h$ | $e$ | $r$ | $I$ | $p$ |
|---|---|---|---|---|---|---|---|---|---|
| Pixel3 Scaling (◆) | **74.8** | 20.91 | **181.07** | 16 | 2 | 108 | 4 | 160 | 16 |
| TX2 Scaling (⬠) | 74.0 (↓ **0.8**) | 14.44 (↓ **30.94%**) | 275.06 (↑ **51.90%**) | **6** | **4** | **64** | **16** | 160 | 16 |
| TX2 Scaling (⬠) | **78.2** | 23.70 | 456.41 | 10 | 4 | 64 | 16 | 160 | 16 |
| Pixel3 Scaling (◆) | 77.5 (↓ **0.7**) | 27.43 (↑ **15.74%**) | 297.58 (↓ **34.80%**) | **16** | **2** | **142** | **4** | 160 | 16 |

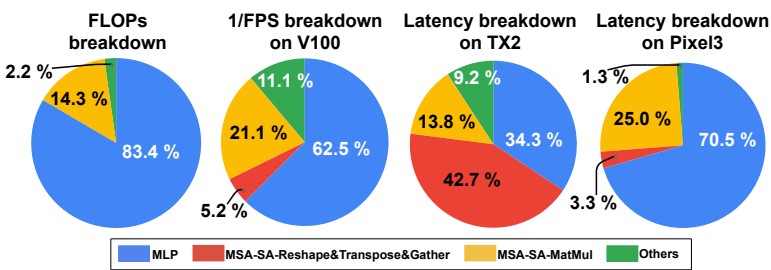

Figure 7: Cost breakdown of DeiT-Tiny on different devices in terms of (1) the number of FLOPs, (2) the 1/FPS on V100, (3) the latency on TX2, and (4) the latency on Pixel3, where MLP represents the cost of all the Linear layers of ViT, MSA-SA-MatMul represents the cost of matrix multiplication among Q(uery), K(ey), and V(alue) in ViT's multi-head attention, MSA-SA-Reshape&Transpose&Gather represents the cost of merely the data movement in ViT's multi-head attention, and the cost of all other operators are denoted as Others.

two perspectives: (1) cost (e.g., latency) breakdown of the same model on different devices and (2) the rank correlation between the hardware-cost on different devices for the same group of models.

**Connection between the breakdown and the transferability effectiveness.** As shown in Figure 7, the cost breakdown of the DeiT-Tiny model suggests that the breakdown in terms of the number of FLOPs, the latency on V100, and the latency on Pixel3 are relatively similar, e.g., the breakdown's cosine distance between any pair among them is smaller than 0.02, while the breakdown for the latency on TX2 is quite different from that of the number of FLOPs, the latency on V100, and the latency on Pixel3, e.g., the breakdown's cosine distance between the latency on TX2 and any other metric is larger than 0.28). This breakdown analysis explains why the scaled models targeting FLOPs (denoted as ☆), V100 (▲), and TX2 (⬟) have a different transferability performance in terms of the accuracy-latency trade-off when executed on Pixel3.

**Rank correlation between the hardware-cost on different devices can also indicate the transferability effectiveness.** Besides the above analysis based on the cost breakdown on different devices using one specific model (i.e., DeiT-Tiny), we also perform analysis based on a group of ViT models.

Following the extensive search adopted in (Bello et al., 2021), we generate a group of ViT models by varying $d$ in [3, 6, 12, 18, 24], $h$ in [2, 3, 6, 8, 12], $e$ in [32, 64, 96], $r$ in [2, 4, 8], $I$ in [128, 160, 224, 320], and $p$ in [8, 16, 32], resulting in a total of **2,700 different ViT models**. As shown in Figure 8, the Kendall Rank Correlation Coefficient Coefficient (Abdi, 2007), which is commonly used to benchmark the effectiveness of accuracy/hardware-cost predictors in recent neural architecture search works (Li et al., 2021; You et al., 2020; Dai et al., 2020), between the latency on Pixel3 and TX2 (highlighted in the red box) is the lowest one among all the coefficients. This set of experiments indicates the weaker performance of using the latency on *TX2/Pixel3* to be the proxy metric when scaling ViT targeting *Pixel3/TX2*, as compared to other device pairs, which is consistent with our observations on the transferability performance among different devices in Section 4.2.1, i.e., scaled models targeting FLOPs (denoted as ☆) and V100 (denoted as ▲) have a better transferability performance when executed on Pixel3 than those targeting TX2 (denoted as ⬟).

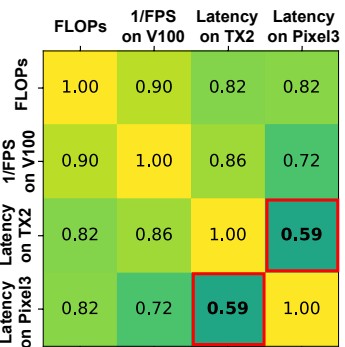

Figure 8: The rank correlation coefficient between the hardware-cost on different devices.

Along with the above analysis based on the (1) cost breakdown and (2) rank correlation between the hardware-cost on different devices, we further perform a deeper analysis from the hardware device specification perspective in Appendix D for better understanding why the transferability effectiveness depends on the underlying devices.

## 4.3 TRANSFER OUR SCALING STRATEGIES ACROSS DIFFERENT MODELS AND TASKS

To answer "*can these scaling strategies be transferred to different ViT variants and tasks?*", we transfer the extracted scaling strategies in Section 4.1 for DeiT (Touvron et al., 2020) on ImageNet (Deng

et al., 2009), as illustrated in Figure 5, to (1) **PiT** (Heo et al., 2021), a strong ViT variant targeting efficiency, on ImageNet (Deng et al., 2009); (2) COCO (Lin et al., 2014), a popular benchmark for **object detection** tasks, to build the backbone of the Deformable DETR detector (Zhu et al., 2020); (3) Kinetics-400 (Kay et al., 2017), a commonly used dataset for **video classification** tasks, with a TimeSFormer (Bertasius et al., 2021) style model extension, which is included in Appendix C.

### 4.3.1 TRANSFER TO THE PiT MODELS

As shown in Table 6, when being transferred to PiT on ImageNet (Deng et al., 2009), the scaling strategies obtained from targeting DeiT (Touvron et al., 2020) on ImageNet (Deng et al., 2009) still lead to advantageous accuracy-efficiency trade-offs for both the PiT-Scaled-Tiny and PiT-Scaled-XS models, e.g., a ↑2.1% and ↑0.4% higher accuracy under a similar number of FLOPs, respectively. Although the accuracy improvement for PiT-Scaled-Small is not as obvious as that for PiT-Scaled-Tiny/XS (i.e., ↑0.1% under similar FLOPs), the transferred scaling strategies at least do not lead to an in-

Table 6: Transferring the scaling strategies targeting DeiT (Touvron et al., 2020) to PiT (Heo et al., 2021), where the resulting models are denoted as PiT-Scaled-Tiny/XS/Small.

| Model | Top-1 accuracy (%) | FLOPs (G) |
|---|---|---|
| PiT-Tiny | 74.6 | 0.71 |
| **PiT-Scaled-Tiny** | **76.7 (↑ 2.1)** | 0.70 |
| PiT-XS | 79.1 | 1.40 |
| **PiT-Scaled-XS** | **79.5 (↑ 0.4)** | 1.38 |
| PiT-Small | 81.9 | 2.9 |
| PiT-Small (Reproduced) | 81.7 | 2.9 |
| **PiT-Scaled-Small** | **81.8 (↑ 0.1)** | 3.0 |

ferior model architecture. More details about the architectures of PiT-Scaled-Tiny/XS/Small are provided in Appendix A.

### 4.3.2 TRANSFER TO AN OBJECT DETECTION TASK

When transferred to object detection, DeiT (Touvron et al., 2020) and our scaled DeiT-Scaled models are inserted into Deformable DETR (Zhu et al., 2020) as the backbones, and the corresponding throughput on V100 is measured using the widely used Detectron2 tool (Wu et al., 2019). As listed in Table 7, our DeiT-Scaled models achieve a ↑0.7% higher average precision under a similar inference throughput, which is consistent with our observation on the advantages of our DeiT-Scaled models over the original DeiT (Touvron et al., 2020) models in terms of classification tasks, which is discussed in Section 4.1.

Table 7: **COCO (Lin et al., 2014) detection** performance (val2017) of DeiT (Touvron et al., 2020) and our DeiT-Scaled models with a Deformable DETR (Zhu et al., 2020) detector.

| Backbone | Average precision (%) | Throughput (FPS) on V100 |
|---|---|---|
| DeiT-Tiny | 35.0 | 13.31 |
| **DeiT-Scaled-Tiny** | **35.7 (↑ 0.7)** | 13.05 |
| DeiT-Small | 41.0 | 10.81 |
| **DeiT-Scaled-Small** | **41.7 (↑ 0.7)** | 9.81 |

All the above attempts of transferring the scaling strategies, extracted from scaling vanilla ViT models on an image classification task, into different ViT variants and tasks share the following common observations: (1) for some cases, such a transfer still achieves advantegeous accuracy-efficiency trade-offs, even without any further exploration of scaling strategies dedicated to the new models/tasks; and (2) for the remaining cases, the transferred scaling strategies lead to models with accuracy-efficiency trade-offs that are on par with the corresponding vanilla models. Notably, there is no extra exploration cost (e.g., re-extracting dedicated scaling strategies) during transfer and thus it can provide at least a good starting point for further dedicated exploration on the new models/tasks.

## 5 CONCLUSION

In this work, we present the study for exploring hardware-aware ViT scaling and show that a simply scaled vanilla ViT model can achieve a comparable or even better (e.g., up to ↑ 3.7% higher accuracy) accuracy-efficiency trade-off as compared to dedicatedly designed SOTA ViT variants. Furthermore, we extract scaling strategies dedicated to ViT and study their transferbility across different hardware devices, ViT variants, and computer vision tasks. We believe that this work has demonstrated a promising perspective towards more efficient/accurate ViT models and will inspire more following innovations on both new ViT models via scaling and hardware-efficient ViT models.

## 6 REPRODUCIBILITY STATEMENT

Regarding our efforts that have been made to ensure reproducibility, we provide the implementation details in Appendix E.

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

## A ARCHITECTURE CONFIGURATIONS OF PiT-SCALED-TINY/XS/SMALL

Table 8: Architecture configuration of PiT-Scaled-Tiny/XS/Small, including image resolution ($I$), spatial size (i.e., # of spatial tokens), # of layers ($d$), # of heads ($h$), and the embedding dimension for each head ($e$). Here, $h$ in the PiT models has to be in $h$-$2h$-$4h$ format (e.g., 2-4-8 in PiT-Tiny).

| Model | FLOPs (G) | Top-1 accuracy (%) | $I$ | Spatial size | $d$ | $h$ | $e$ |
|---|---|---|---|---|---|---|---|
| DeiT-Tiny | 1.26 | 74.5 | 224 | $14 \times 14$ | 12 | 3 | 64 |
| **DeiT-Scaled-Tiny** | 1.22 | **76.4 (↑ 1.9)** | **160** | **$10 \times 10$** | **14** | **4** | 64 |
| PiT-Tiny | 0.71 | 74.6 | 224 | $27 \times 27$ | 2 | 2 | 32 |
| | | | | $14 \times 14$ | 6 | 4 | 32 |
| | | | | $7 \times 7$ | 4 | 8 | 32 |
| **PiT-Scaled-Tiny** | 0.70 | **76.7 (↑ 2.1)** | **160** | **$19 \times 19$** | **2** | **3** | 32 |
| | | | | **$10 \times 10$** | **7** | **6** | 32 |
| | | | | **$5 \times 5$** | **4** | **12** | 32 |
| PiT-XS | 1.41 | 79.1 | 224 | $27 \times 27$ | 2 | 2 | 48 |
| | | | | $14 \times 14$ | 6 | 4 | 48 |
| | | | | $7 \times 7$ | 4 | 8 | 48 |
| **PiT-Scaled-XS** | 1.38 | **79.5 (↑ 0.4)** | **160** | **$19 \times 19$** | **2** | **3** | 48 |
| | | | | **$10 \times 10$** | **6** | **6** | 48 |
| | | | | **$5 \times 5$** | **4** | **12** | 48 |
| DeiT-Small | 4.62 | 81.2 | 224 | $14 \times 14$ | 12 | 6 | 64 |
| **DeiT-Scaled-Small** | 4.79 | **81.6 (↑ 0.4)** | **256** | **$16 \times 16$** | **20** | **4** | 64 |
| PiT-Small | 2.90 | 81.7 | 224 | $27 \times 27$ | 2 | 3 | 48 |
| | | | | $14 \times 14$ | 6 | 6 | 48 |
| | | | | $7 \times 7$ | 4 | 12 | 48 |
| **PiT-Scaled-Small** | 3.04 | **81.8 (↑ 0.1)** | **256** | **$31 \times 31$** | **3** | **2** | 48 |
| | | | | **$16 \times 16$** | **10** | **4** | 48 |
| | | | | **$8 \times 8$** | **6** | **8** | 48 |

Here we provide more details regarding how we transfer our extracted strategies to other ViT models, e.g., the PiT models. Specifically, to obtain the corresponding PiT-Scaled-Tiny/XS/Small models based on the baseline PiT-Tiny/XS/Small models and extracted scaling strategies, we 1) locate the most suitable architecture configuration in our scaling strategies to be used in the new variant, i.e., DeiT-Scaled-Tiny corresponds to PiT-Tiny/XS, and DeiT-Scaled-Small corresponds to PiT-Small, considering that PiT-Tiny/XS/Small are designed to be at a scale similar to DeiT-Tiny/Small (Heo et al., 2021)); 2) adjust the scaling factors which are the same in both DeiT and the new variant baseline models to match the located architecture configuration in the previous step, e.g., adjusting $I$ from 224 to 160 in PiT-Tiny to build PiT-Scaled-Tiny; and 3) scale down/up the remaining scaling factors if the transferred models cost more/less FLOPs than the new variant baseline models, e.g., scaling up $h$ from 2-4-8 to 3-6-12 in PiT-Tiny to build PiT-Scaled-Tiny. The details of the finally obtained PiT-Scaled-Tiny/XS/Small models are summarized in Table 8.

## B DEVICES SETUP

### B.1 NVIDIA V100

**Device specifications and target applications.** NVIDIA V100 (V100) (NVIDIA LLC.) is one of the most advanced data center GPUs that accelerate deep learning applications for cloud services and powered by 5120 NVIDIA CUDA cores and 640 NVIDIA Tensor cores. In all our experiments, we use the 16GB HBM2 GPU memory configuration type V100.

**Pre-measurement setup.** The V100 GPU system consists of an Intel Xeon Bronze 3204 Processor and 21GB RAM that are able to provide a high processing throughput (i.e., frames per second) of the given DNN models.

**Measurement pipeline**. Following (Touvron et al., 2020), we use the maximum power-of-two batch size that can fit in the memory when measuring the throughput with the officially provided PyTorch profiler (Maxim Lukiyanov, Guoliang Hua, Geeta Chauhan, and Gisle Dankel) based on on the PyTorch scripts provided in (Francisco Massa).

### B.2 NVIDIA EDGE GPU TX2

**Device specifications and target applications.** NVIDIA Edge GPU TX2 (TX2) (NVIDIA Inc., c) consists of a quad-core Arm Cortex-A57, a dual-core NVIDIA Denver2, a 256-core Pascal GPU, and a 8GB 128-bit LPDDR4. It is commonly used in IoT and self-driving environments (Li et al., 2020; Siam et al., 2018; Wofk et al., 2019), working as an edge computing platform with a relatively weak GPU.

**Pre-measurement setup.** In order to make full use of its resource following (Wofk et al., 2019), we enable jetson_clock (NVIDIA Inc., a) on TX2, pre-setting it into a max-N mode and adjusting the fan speed to 100%.

**Measurement pipeline**. When we measure the latency of a specific model on TX2, the model definition in PyTorch (Paszke et al., 2019) will be 1) exported into the onnx format (Bai et al., 2019) and 2) passed to the TensorRT command-line wrapper (NVIDIA Inc., d), an officially provided binary file, to be executed by TensorRT (NVIDIA Inc., b) that is a C++ library for high-performance inference on NVIDIA GPUs. The corresponding latency is directly reported by the TensorRT command-line wrapper (NVIDIA Inc., d).

### B.3 GOOGLE PIXEL3

**Device specifications and target applications.** Google Pixel3 (Pixel3) (Google LLC., a) consists of a quad-core 2.5 GHz Kryo 385 Gold CPU, a quad-core 1.6 GHz Kryo 385 Silver CPU, and a 4GB RAM. It is one of the latest Pixel mobile phones, which is widely used as the benchmark platform for deep learning targeting mobile devices (Xiong et al., 2020; Howard et al., 2019; Google LLC., c).

**Pre-measurement setup.** In order to reduce the variance of the measured latency, the Pixel3 device is pre-configured to only use its big cores to perform the network inference, following the settings in (Xiong et al., 2020; Google LLC., b).

**Measurement pipeline**. To operate a given model in Pixel3, the model will be 1) converted into the tflite format (Google LLC., c) and 2) passed to the tflite benchmark tools (Google LLC., b) that are an officially provided binary file for fairly benchmarking different models in tflite. The corresponding latency is then directly reported by the tflite benchmark tools (Google LLC., b).

## C TRANSFER OUR SCALING STRATEGIES TO VIDEO CLASSIFICATION TASKS

When transferring our scaling strategies to video classification tasks, we follow (Bertasius et al., 2021) to (1) decompose an input video into a sequence of frame-level patches and feed them into a Transformer module and (2) include two attention schemes, "Joint" (i.e., applying self-attention into space-time tokens jointly) and "Divided" (i.e., applying spatial and temporal attentions separately), to benchmark the performance of different models. As shown in Table 9, our DeiT-Scaled models (e.g. DeiT-Scaled-Tiny) can reduce the FLOPs by 33.2% under a similar accuracy (67.4% vs 67.7%) as compared to DeiT-Tiny with the "Joint" attention scheme, and achieve accuracy-FLOPs trade-offs at least no worse than the original DeiT (Touvron et al., 2020) models in other settings.

Table 9: **Kinetics-400 (Kay et al., 2017) video classification** performance (validation set) of extended DeiT (Touvron et al., 2020) and our DeiT-Scaled models with a TimeSFormer (Bertasius et al., 2021) style.

| Attention Scheme | Model | Top-1 Accuracy (%) | FLOPs (G) |
|---|---|---|---|
| Joint | DeiT-Tiny | 67.7 | 19.9 |
| | **DeiT-Scaled-Tiny** | 67.4 (↓ **0.3**) | 13.3 (↓ **33.2%**) |
| | DeiT-Small | 71.2 | 56.5 |
| | **DeiT-Scaled-Small** | 71.4 (↑ **0.2**) | 61.9 (↑ **9.56%**) |
| Divided | DeiT-Tiny | 68.4 | 13.6 |
| | **DeiT-Scaled-Tiny** | 67.8 (↓ **0.6**) | 12.7 (↓ **6.62%**) |
| | DeiT-Small | 71.4 | 50.8 |
| | **DeiT-Scaled-Small** | 72.0 (↑ **0.6**) | 54.2 (↑ **6.69%**) |

Table 10: Specifications of the hardware devices in the transferability exploration experiments.

| Specifications | NVIDIA V100 System (V100) | NVIDIA Edge GPU TX2 (TX2) | Google Pixel3 (Pixel3) |
|---|---|---|---|
| GPU Architecture | NVIDIA Volta | NVIDIA Pascal | Qualcomm Adreno |
| CUDA Cores | 5120 | 256 | - |
| CPU | AMD EPYC 7742 | NVIDIA Denver 2/ARM® Cortex®-A57 | Kryo 385 Gold/Kryo 385 Silver |
| CPU Max Frequency | 3.4 GHz | 2 GHz/2 GHz | 2.8 GHz/1.7 GHz |
| GPU/SoC Memory | 16 GB | 8 GB | 4 GB |
| Power Consumption | 300 W | 15 W | 18 W |

Table 11: Detailed cost breakdown of DeiT-Tiny on different devices for the operators of (1) multi-layer perceptron (MLP), (2) layer normalization (LayerNorm), (3) matrix multiplication in multi-head self-attention (MSA-MatMul), (4) softmax in multi-head self-attention (MSA-Softmax), (5) reshape and transpose in multi-head self-attention (MSA-Reshape&Transpose), (6) gather in multi-head self-attention (MSA-Gather), and (7) others.

| Operators | 1/FPS on V100 (%) | Latency on TX2 (%) | Latency on Pixel3 (%) |
|---|---|---|---|
| **MLP** | **62.50** | **34.31** | **69.40** |
| LayerNorm | 8.95 | 6.38 | 1.59 |
| MSA-MatMul | 17.65 | 8.03 | 21.36 |
| MSA-Softmax | 3.48 | 5.75 | 3.20 |
| MSA-Reshape&Transpose | 5.17 | 6.32 | 3.30 |
| **MSA-Gather** | **<0.01** | **36.38** | **<0.01** |
| Others | 2.20 | 2.82 | 1.26 |

# D   ANALYSIS ON THE TRANSFERABILITY ACROSS DIFFERENT DEVICES FROM THE HARDWARE DEVICE SPECIFICATIONS PERSPECTIVE

By observing the specifications of different hardware devices, which is summarized in Table 10, and the detailed cost breakdown on different devices in Table 11, we can conclude that (1) the most significant differences come from the MLP and MSA-Gather operators for all the three devices, e.g., MSA-Gather costs much more (36.38% vs. <0.01%) and MLP costs much less (34.31% vs. 62.50%/69.40%) in TX2 than in Pixel3/V100 and (2) TX2 has the weakest CPU in terms of the maximum frequency among the three devices. Thus, we conjecture the slow data movements in TX2 due to the weakest CPU cause the largest MSA-Gather cost percentage in TX2 among these devices. This can explain that the scaling strategies obtained when targeting FLOPs and V100 can be transferred to Pixel3 with little or even no performance loss,but those obtained for TX2 cannot do that, as mentioned in Section 4.2.

Interestingly, by comparing the extracted scaling strategies for V100 and TX2, we can observe that the scaled ViT in TX2 tends to enlarge more on linear projection ratio ($r$), which will not increase the cost of self-attention, as compared to the scaled ViT in V100 (16 vs. 4) under a similar accuracy (78.17% vs. 78.10%), as shown in Table 12. This matches the observation that the self-attention costs a large portion of the cost on TX2 (e.g., 56.48% for DeiT-Tiny), as demonstrated in Table 11.

Table 12: Detailed architecture configurations of the scaled ViT models with throughput (i.e., FPS) on V100 (V100 Scaling) and latency on TX2 (TX2 Scaling) as the hardware-cost during scaling, respectively.

| Metrics | V100 Scaling | TX2 Scaling |
|---|---|---|
| Accuracy (%) | 78.10 | 78.17 |
| **FPS on V100** | **2488.81** | 1984.10 |
| **Latency on TX2 (ms)** | 25.18 | **23.70** |
| Num. of layers ($d$) | 13 | 10 |
| Num. of heads ($h$) | 5 | 4 |
| Embedding dim. per head ($e$) | 64 | 64 |
| **Linear projection ratio ($r$)** | **4** | **16** |
| Image resolution ($I$) | 160 | 160 |
| Patch size ($p$) | 16 | 16 |

## E    IMPLEMENTATION DETAILS

In this section, we provide the implementation details of our experiments, including (1) our scaled ViT (Dosovitskiy et al., 2021; Touvron et al., 2020) models on ImageNet (Deng et al., 2009) dataset in Section 4.1 and 4.2, (2) our scaled PiT (Heo et al., 2021) models on ImageNet (Deng et al., 2009) dataset in Section 4.3.1, (3) our scaled ViT (Dosovitskiy et al., 2021; Touvron et al., 2020) models on COCO (Lin et al., 2014) dataset in Section 4.3.2, and (4) our scaled ViT (Dosovitskiy et al., 2021; Touvron et al., 2020) models on Kinetics-400 (Kay et al., 2017) dataset in Appendix C.

**Scaled ViT models on ImageNet dataset.** All the scaled ViT (Dosovitskiy et al., 2021; Touvron et al., 2020) models on ImageNet (Deng et al., 2009) dataset reported in Section 4.1 and 4.2 follow the same training recipe (including the data pre-processing) with the one proposed in (Touvron et al., 2020), i.e., training on ImageNet for 300 epochs (1000 epochs for models in Table 3) with batch size as 1024, AdamW optimizer (Loshchilov & Hutter, 2017), learning rate as 0.001, cosine learning rate decay, weight decay as 0.05, 5 warmup epochs, and distillation from RegNetY-16GF (Radosavovic et al., 2020).

**Scaled PiT models on ImageNet dataset.** To make a fair comparison with PiT (Heo et al., 2021) models, all our scaled PiT models (i.e., PiT-Scaled-Tiny/XS/Small in Table 6) follow the training recipe (including the data pre-processing) in PiT (Heo et al., 2021), which uses the same learning rate, weight decay, warmup epochs, total epochs, and distillation settings with (Touvron et al., 2020), but using AdamP (Heo et al., 2020) as the optimizer instead of AdamW (Loshchilov & Hutter, 2017).

**Scaled ViT models on COCO dataset.** Following the training recipe (including the data pre-processing) described in (Zhu et al., 2020), all the models are pre-trained on ImageNet (Deng et al., 2009) first, and then trained on COCO (Lin et al., 2014) dataset for 50 epochs with Adam optimizer, learning rate as 0.0002, weight decay 0.0001, and the learning rate is decayed at the 40-th epoch by a factor of 0.1. Note that when adapting the models pre-trained on ImageNet(Deng et al., 2009) to COCO (Lin et al., 2014), we scale the positional embeddings of ViT via bilinear interpolation to match the differences of image resolutions and use the feature map before the final classifier and layernorm layer as the input feature map to the Deformable DETR header.

**Scaled ViT models on Kinetics-400 dataset.** For Kinetics-400 (Kay et al., 2017) dataset, we follow the training recipes (including the data pre-processing) in (Bertasius et al., 2021) to start from the ImageNet (Deng et al., 2009) pre-trained models. Then clips of size $8 \times 224 \times 224$ with frames sampled as a rate of 1/32 are used for training. All models are trained for 15 epochs with learning rate as 0.005, batch size as 16, SGD optimizer with momentum 0.9, and the learning rate is decayed at the 10-th and 14-th epoch by a factor of 0.1. We also include both the "Joint" (i.e., applying self-attention into space-time tokens jointly) and "Divided" (i.e., applying spatial and temporal attentions separately) attention schemes described in (Bertasius et al., 2021) to make a more fair comparison with the baseline models which use DeiT (Touvron et al., 2020) models as backbones.

## F    ARCHITECTURES COMPARISON BETWEEN THE SCALED MODELS AND THE RANDOMLY PERMUTATED MODELS

To further explore why the architectures with the best accuracy vs. efficiency trade-off after the random permutation on top the scaled models can achieve better performance than the scaled models, as shown in Figure 4 of the main content, here we summarize their performance and architectures details in Table 13. As compared to the scaled models (i.e., DeiT-Scale-{Tiny, Small}), those architectures with the best accuraccy vs. efficiency trade-off after random permutation (i.e., DeiT-Scale-{Tiny, Small}-RP) adopt different scaling factors except the number of heads.

Table 13: Random permutation further boosts the performance of the scaled models.

| Model | FLOPs (G) | Top-1 accuracy (%) | $d$ | $h$ | $e$ | $r$ | $I$ | $p$ |
|---|---|---|---|---|---|---|---|---|
| DeiT-Tiny | 1.26 | 74.5 | 12 | 3 | 64 | 4 | 224 | 16 |
| **DeiT-Scaled-Tiny** | 1.22 | **76.4 (↑1.9)** | **14** | **4** | **64** | **4** | **160** | **16** |
| **DeiT-Scaled-Tiny-RP** | 1.22 | **76.9 (↑2.4)** | **17** | **4** | **60** | **5** | **171** | **19** |
| DeiT-Small | 4.62 | 81.2 | 12 | 6 | 64 | 4 | 224 | 16 |
| **DeiT-Scaled-Small** | 4.79 | **81.6 (↑0.4)** | **20** | **4** | **64** | **4** | **256** | **16** |
| **DeiT-Scaled-Small-RP** | 4.79 | **82.0 (↑0.8)** | **21** | **4** | **68** | **5** | **210** | **15** |

## G  MORE DETAILS ON THE ITERATIVE GREEDY SEARCH

To better demonstrate the iterative greedy search method introduced in Section 3.3, we summarize the important details in Algorithm 1. In a nutshell, we 1) start from a small model defined in Table 1; 2) in each scaling step, we increase each scaling factor alone to meet the pre-set hardware-cost target; 3) we then select the one with the best accuracy vs. efficiency trade-off out of those architectures resulting from increasing each scaling factor standalone in the previous step; and 4) the selected architecture from the previous step will be used as the starting point in the next step.

---

**Algorithm 1** Iterative Greedy Search

1: Step: $s \leftarrow 0$
2: Architecture: $A \leftarrow A_0$ ($A_0$ is shown in Table 1)
3: Hardware-cost metric: $C()$
4: Total steps in the scaling: $N$
5: Target hardware-cost in each step: $[C_1, C_2, ..., C_N]$
6: **for** $s$ in $[1, 2, ..., N]$ **do**
7:     **for** scaling factor $SF$ in $[d, h, e, r, I, p]$ **do**
8:         $A_{SF} \leftarrow A$
9:         **while** $C(A_{SF}) < C_s$ **do**
10:           $SF ++$
11:         **end while**
12:         $A^s_{SF} \leftarrow A_{SF}$
13:     **end for**
14:     $A^s \leftarrow \max\_accuracy(A^s_d, A^s_h, A^s_e, A^s_r, A^s_I, A^s_p)$
15:     $A \leftarrow A^s$
16: **end for**
17: Searched architectures $A_1, A_2, ..., A_N$

---

