# OpenReview forum: "An Investigation on Hardware-Aware Vision Transformer Scaling"
_ICLR.cc/2022/Conference — ICLR 2022 Submitted_

### Official Review · Reviewer_WJMX · 2021-11-02

**Correctness:** 3
**Technical Novelty And Significance:** 3
**Empirical Novelty And Significance:** 4
**Recommendation:** 6
**Confidence:** 5

**Main Review:**

Strengths:
1.This work provides a new perspective to explore ViT architectures, aiming to analyze and understander ViT’s architecture design space and real hardware-cost across different devices.
2. This work summarizes how to determine if a model searched on one hardware device applies to another.

Weakness:
1.Some typos such as “TRAFE-OFFS” in the title of section 4.1.
2.The 24 different structures generated by random premutation in section 4.1 should be explained in more detail.
3.The penultimate sentence of Section 3.3 states that "iterative greedy search can avoid the suboptimality of the resulting scaling strategy on a particular model", which is not a serious statement because the results of the iterative greedy search are also suboptimal solutions.
4.The conclusion of "Cost breakdown can indicate the transferability effectiveness" in Figure 7 is not sufficient. We cannot extend the conclusion obtained from a few specific experiments to any different hardware devices or different architectures.
5.Why not use the same cost for different devices instead of flops, latency, and 1/FPS for different hardware?
6.The result comparison of "Iteratively greedy Search" versus "random search" on the model structure should be supplemented.


**Summary Of The Paper:**

This work presents a study for exploring hardware-aware ViT scaling and demonstrates that a simply scaled vanilla ViT model can achieve a comparable accuracy-efficiency trade-off as compared to dedicatedly designed ViT variants. Furthermore, they discuss the transferability of the scaling strategies on different hardware devices, ViT variants, and computer vision tasks.

**Summary Of The Review:**

Please provide a short summary justifying your recommendation of the paper
Although some parts of the paper are not clearly explained and the method is not very innovative, the analysis of the transferability of different scaling strategies on different hardware devices in this work is very positive and provides great inspiration to relevant fields. Therefore, it is recommended to accept this paper.

---

> ### Author Response · Authors · 2021-11-23
> **Response to Reviewer WJMX**
>
> We greatly appreciate your careful reading and deep understanding of our work and motivation, which is as you pointed out “the analysis of the transferability of different scaling strategies on different hardware devices in this work is very positive and provides great inspiration to relevant fields”.
>
> > Q1: Some typos such as “TRAFE-OFFS” in the title of section 4.1
>
> A1: Thank you for pointing it out! We have revised it in the updated manuscript and will proofread it more carefully in our final version.
>
> > Q2: The 24 different structures generated by random permutation in section 4.1 should be explained in more detail.
>
> A2: Thank you for your suggestions! The random permutation is implemented by randomly mutating the scaling factors (i.e., the number of layers, number of heads, embedding dimension per head, linear projection ratio, image resolution, and patch size) of the scaled architectures that are obtained by first performing iterative greedy search and then selecting those under the target hardware-cost. We have also added more details to the corresponding part for further clarification in Section 4.1.
>
> > Q3: The penultimate sentence of Section 3.3 states that "iterative greedy search can avoid the suboptimality of the resulting scaling strategy on a particular model", which is not a serious statement because the results of the iterative greedy search are also suboptimal solutions.
>
> A3: Thank you for pointing it out! We have changed the corresponding claims in Section 3.3 of the updated manuscript.
>
>
> > Q4: The conclusion of "Cost breakdown can indicate the transferability effectiveness" in Figure 7 is not sufficient. We cannot extend the conclusion obtained from a few specific experiments to any different hardware devices or different architectures.
>
> A4: Thank you for your comment! We did have a consistent observation across different ViT models and only took the case of DeiT-Tiny as an illustrative example in our submitted manuscript. We summarize the observation on more ViT models in the table below, from which we can see that different models share the same observation: the breakdown in terms of the number of FLOPs, the latency on V100, and the latency on Pixel3 are relatively similar, while the breakdown for the latency on TX2 is quite different from others, especially on the Reshape&Transpose&Gather operators highlighted in bold in the tabled below. This matches the experiments in Section 4.2.1 of our submitted manuscript, which indicate that the scaled models targeting different types of hardware-cost have a different transferability performance in terms of accuracy-latency trade-off when executed on Pixel3. Also, to avoid misunderstanding, we have changed the claim of "cost breakdown can indicate the transferability effectiveness" to "connection between the breakdown and the transferability effectiveness" in the updated manuscript.
>
>
> | Model| | DeiT-Tiny | DeiT-Small | DeiT-Base|
> |:-:|:-:|:-:|:-:|:-:|
> | |MLP| 83.4 | 90.9 | 95.2|
> | FLOPs |MSA-SA-Reshape&Transpose&Gather | 0.0 | 0.0 | 0.0|
> | breakdown |MSA-SA-MatMul | 14.3 | 7.8 | 4.1 |
> | |Others | 2.2 | 1.3 | 0.7 |
> | - | - | - | - | - |
> | |MLP| 62.5 | 74.0 | 83.8 |
> | 1/FPS breakdown |MSA-SA-Reshape&Transpose&Gather | 5.2 | 3.9 | 2.6|
> | on V100 |MSA-SA-MatMul | 21.1 | 16.2 | 10.4|
> | |Others | 11.1 | 5.9 | 3.2|
> | - | - | - | - | - |
> | |MLP| 34.3 | 51.1 | 63.5|
> | **Latency breakdown** | **MSA-SA-Reshape&Transpose&Gather** | **42.7** | **29.6** | **22.3** |
> | **on TX2** |MSA-SA-MatMul | 13.8 | 12.5 | 9.3|
> | |Others | 9.2 | 6.8 | 4.9|
> | - | - | - | - | - |
> | |MLP|70.5 | 78.3 | 86.8|
> | Latency breakdown | MSA-SA-Reshape&Transpose&Gather | 3.3 | 2.9 | 1.8|
> | on Pixel3 |MSA-SA-MatMul | 25.0 | 17.8 | 11.2|
> | |Others | 1.3 | 0.5 | 0.1|
>
>
> > Q5: Why not use the same cost for different devices instead of flops, latency, and 1/FPS for different hardware?
>
>
> A5: We adopt the commonly used hardware-cost metric for each device. For Pixel 3 and TX2, latency measured with batch size 1 is a widely used setting [1,2]; for V100, the throughput (i.e., FPS) under the max batch size to fill the GPU memory is commonly used to benchmark different models’ performance [3,4]. Furthermore, to make the adopted metric have a consistent meaning, i.e., the higher the more costly, we use latency for Pixel 3&TX2 and 1/FPS for V100.
>
> > + Q6: The result comparison of "Iteratively greedy Search" versus "random search" on the model structure should be supplemented.
>
> A6: Thanks for your suggestions! We have included the architecture comparison between those architectures scaled by the iteratively greedy search and those randomly permuted on top of the scaled models in Appendix F of our updated manuscript.

---

> > ### Author Response · Authors · 2021-11-23
> > **Response to Reviewer WJMX**
> >
> >
> > *[1] Xiong, Yunyang, et al. "Mobiledets: Searching for object detection architectures for mobile accelerators." Proceedings of the IEEE/CVF Conference on Computer Vision and Pattern Recognition. 2021.*
> >
> > *[2] Li, Chaojian, et al. "Hw-nas-bench: Hardware-aware neural architecture search benchmark." arXiv preprint arXiv:2103.10584 (2021).*
> >
> > *[3] Touvron, Hugo, et al. "Training data-efficient image transformers & distillation through attention." International Conference on Machine Learning. PMLR, 2021.*
> >
> > *[4] Graham, Ben, et al. "LeViT: a Vision Transformer in ConvNet's Clothing for Faster Inference." arXiv preprint arXiv:2104.01136 (2021).*

---

### Official Review · Reviewer_Lm1r · 2021-11-02

**Correctness:** 3
**Technical Novelty And Significance:** 2
**Empirical Novelty And Significance:** 3
**Recommendation:** 5
**Confidence:** 3

**Main Review:**

Strength:

-This paper analyzes different scaling strategies of vision transformers and proposes a hard-ware scaling strategy for vision transformers. The authors conduct extensive experiments and bring some insights.

-The proposed model achieves a better trade-off between FLOPs and accuracies than the existing models. It is interesting to see that only adjusting the model size can achieve better performance than SOTA transformer models.

Weakness:
- [1] has discussed how to scale vision transformers. A more detailed comparison with [1] is required.

- Hardware-Aware architecture searching has been well investigated. The proposed method is not novel.

[1] Scaling Vision Transformers, arxiv 2106




**Summary Of The Paper:**

This paper proposes a model scaling strategy to change the model size. With a greedy search strategy, the author proposes a hardware-aware scaling method. Compared with the existing ViT variants, the proposed method achieves better performance.

**Summary Of The Review:**

The proposed method is not novel, but the insights brought by the authors are interesting.

---

> ### Author Response · Authors · 2021-11-23
> **Response to Reviewer Lm1r**
>
> > + Q1: [1] has discussed how to scale vision transformers. A more detailed comparison with [1] is required.
>
> A1: Thank you for pointing this out! We humbly remind that we have discussed the comparison with [1] in Section 2 of our submitted manuscript, in case you missed it. To make it more clear, we further listed the main differences between our work and [1] as follows: 1) we focus more on the accuracy vs. efficiency trade-off when scaling ViTs, while [1] cares more about merely the accuracy; 2) we provide extra analysis on the transferability of the extracted scaling strategies across different devices, ViT variants, and tasks, while [1] only includes the experiments of the transferability to few-shot learning tasks. The corresponding parts in Section 2 have also been modified for further clarification in the updated manuscript.
>
> > + Q2: Hardware-Aware architecture searching has been well investigated. The proposed method is not novel.
>
> A2:  We hope to humbly clarify that we did not  claim the novelty of our search method. Instead, our main contributions include the study and analysis of the scaled architectures and their corresponding transferability across different devices, ViT variants, and recognition tasks. Furthermore, as you pointed out, we show that “only adjusting the model size can achieve better performance than SOTA transformer models”, which can provide a new perspective for designing ViTs with more favorable accuracy-efficiency trade-offs and can shed light on further innovations. Finally, we hope to humbly remind that our work’s contributions, including your mentioned “The contributions are significant and somewhat new” and “It is interesting to see that only adjusting the model size can achieve better performance than SOTA transformer models”, match the ICLR criterion of  “bring value to the ICLR community when they convincingly demonstrate new, relevant, impactful knowledge” mentioned in the ICLR Reviewer Guide. The “usefulness, practical friendliness, and generalization of the finding” of our work have also been recognized by Reviewer zFw4, and Reviewer WJMX commented our work with “This work provides a new perspective to explore ViT architectures”.
>
>
> *[1] Zhai, Xiaohua, et al. "Scaling vision transformers." arXiv preprint arXiv:2106.04560 (2021).*

---

### Official Review · Reviewer_d8q8 · 2021-11-05

**Correctness:** 4
**Technical Novelty And Significance:** 2
**Empirical Novelty And Significance:** 3
**Recommendation:** 3
**Confidence:** 3

**Main Review:**

Strength:
- Approach is tested on multiple devices under different constrains (FLOPs, latency etc). It is clearly shown that optimization for a specific hardware results in a better corresponding model.
- Scaling factors for models are clearly indicated in the tables for reproducibility.
- Improvements are shown over the baseline Deit models and are up to 1.9% better.

Weakness:
- For Figure 1. Did authors include all recent transformer models? For example LeViT is not there, it demonstrates higher accuracy and lower number of FLOPs (LeViT is 1.2 and 2.4x faster than PiT). If it holds then authors should be careful when comparing to hand designed ViT and claiming comparable or better accuracy-efficiency trade-off. Will be interesting to look at the updated figure with recent models.
- Scaling method based on an iterative greedy search is not presented well. Going over the section 3.3 multiple times, it is still not clear how exactly the search was performed. Additionally, it is not obvious how different target constraints are reached (FLOPs or latency).

Minor:
- In the introduction when comparing to CNNs, author mention EfficientNetV1, however, a much stronger EfficientNetV2 has been published.
- Include top performing CNN models into Figure 1.
- Existing calling methods for ViT. in section 3.2 authors first talk about CNN scaling strategies and conclude that those applied to ViTs can lead to ambiguity and sub-optimal performance. Where can we find supporting evidence for the statement?


**Summary Of The Paper:**

The paper is focused on scaling width, input image resolution and other configurations of ViT model (6 in total) to make it hardware friendly and reach better accuracy when trained from the scratch. Results are verified on the edge devices - Nvidia TX2, Pixel3. Models optimized for different hardware result in higher performance on the targeted device.


**Summary Of The Review:**

Paper contains an interesting empirical study on how iterative greedy search can be applied to the design parameters of ViT. Resulting models show speed-latency improvements over hand designed ViT models. The method of the search is not well explained (it seems to be the previously proposed method). Resulting models seem to be less generic and more engineered, not clear how presented results will scale beyond Tiny/Small/Base models. Presented study will be interesting for people designing ViT models, however its limitation is in the conclusions/findings specific only to ViT models (even with transfer to PiT).

---

> ### Author Response · Authors · 2021-11-23
> **Response to Reviewer d8q8**
>
> > + Q1: For Figure 1. Did authors include all recent transformer models? For example LeViT is not there, it demonstrates higher accuracy and lower number of FLOPs (LeViT is 1.2 and 2.4x faster than PiT). If it holds then authors should be careful when comparing to hand designed ViT and claiming comparable or better accuracy-efficiency trade-off. Will be interesting to look at the updated figure with recent models.
>
> A1: Thank you for your suggestions! As we all recognize, the field of ViTs is evolving very fast now, and thus we were not able to include all the recent transformer models. Meanwhile, we hope to humbly remind that ICLR doesn’t require submissions to compare with works published on or after June 5, 2021 as mentioned in the Reviewer Guide, so we did not include all recent transformer models, e.g., LeViT [1] published in ICCV’21 which released the decision to authors on July 22, 2021. Following your suggestion, we have modified the corresponding claim about Figure 1 in our updated manuscript.
>
> > + Q2: Scaling method based on an iterative greedy search is not presented well. Going over the section 3.3 multiple times, it is still not clear how exactly the search was performed. Additionally, it is not obvious how different target constraints are reached (FLOPs or latency).
>
> A2: Sorry for not making it sufficiently clear. To better demonstrate how the search is performed and how the target constraints are reached, we have added the details of our iterative greedy search into the algorithm illustrated in Appendix G. In a nutshell, we 1) start from a small model defined in Table 1; 2) in each scaling step, we increase each scaling factor alone to meet the pre-set hardware-cost target; 3) we then select the one with the best accuracy vs. efficiency trade-off out of those architectures resulting from increasing each scaling factor standalone in the previous step; and 4) the selected architecture from the previous step will be used as the starting point in the next step. Regarding how different target constraints are reached, we gradually increase the scaling factors until the corresponding architecture’s hardware-cost is larger than the target one (i.e., line 9 - 12 in Algorithm 1 of Appendix G).
>
> > + Q3: In the introduction when comparing to CNNs, author mention EfficientNetV1, however, a much stronger EfficientNetV2 has been published.
>
> A3: Thank you for your suggestion! We have changed the corresponding description to EfficienNetV1 in our revised manuscript.
>
>
> > + Q4: Include top performing CNN models into Figure 1.
>
> A4: Thank you for your suggestion! Including the top performing CNN models into Figure 1 may not be fair because we focus on scaling ViT in this work. Current top performing ViT models still have an obvious gap as compared to top performing CNN models, in terms of the accuracy vs. efficiency trade-off, as shown in the table below.
>
>
> | Model | Type | FLOPs (M) | ImageNet Top-1 Accuracy (%) |
> |:-:|:-:|:-:|:-:|
> |LeViT-128 [1] | ViT | 406 | 78.6|
> |Mobile-Former-508M [3] | ViT-CNN hybrid | 508 | 79.3|
> |**FBNetV3-A [2]** | **CNN** | **357** | **79.6** |
> |LeViT-192 [1] | ViT | 658 | 80.0|
> |**FBNetV3-C [2]** | **CNN** | **557** | **80.8** |
> |LeViT-256 [1] | ViT | 1120 | 81.6|
> | **FBNetV3-F [2]** | **CNN** | **1182** | **82.5** |
>
>
> > + Q5: Existing calling methods for ViT. in section 3.2 authors first talk about CNN scaling strategies and conclude that those applied to ViTs can lead to ambiguity and sub-optimal performance. Where can we find supporting evidence for the statement?
>
> A5: Good question! One example is the extracted scaling strategies for TX2, as shown in Appendix D. Specifically, we can observe that the scaled ViTs in TX2 tend to enlarge more on the linear projection ratio ($r$), while the linear projection ratio does not exist in the scaling factors for CNNs. So applying the CNN scaling strategies to ViT while ignoring the linear projection ratio will lead to a sub-optimal performance in terms of accuracy vs. latency on TX2 trade-off. To make it more clear, we have also modified the corresponding description in Section 3.2 in the updated manuscript.
>
> **[1] Graham, Ben, et al. "LeViT: a Vision Transformer in ConvNet's Clothing for Faster Inference." arXiv preprint arXiv:2104.01136 (2021).**
>
> **[2] Dai, Xiaoliang, et al. "FBNetV3: Joint Architecture-Recipe Search using Predictor Pretraining." Proceedings of the IEEE/CVF Conference on Computer Vision and Pattern Recognition. 2021.**
>
> **[3] Chen, Yinpeng, et al. "Mobile-Former: Bridging MobileNet and Transformer." arXiv preprint arXiv:2108.05895 (2021).**

---

> > ### Comment · Reviewer_d8q8 · 2021-11-24
> > **Feedback on rebuttal**
> >
> > Thank you for providing the response. Taking it into account and other reviews, my score is lowered.
> >
> > Main reasons for lowering the score:
> > 1) We see no comparison for LeViT that was available on arXiv in April, that is 6 month old enough (in my understanding
> > mentioned policy doesn't exclude ArXiv). It is not crucial, however, could make the paper stronger.
> > 2) The method of scaling up models is very expensive (retraining is required) and not novel. Therefore contribution is strictly experimental.
> > 3) The evidence of CNN scaling strategies is not addressed, as it is clear that it should be changed according the parameters of the architecture.
> > 4) Including CNN models into Fig 1 would be beneficial to see CNN models that are outperformed by the proposed scaling, being better than all CNN is not important.
> > 5) Paper has useful experimental findings and agree with the reviewer zFw4 that paper is more on the technical report side and would be a great fit for a workshop.

---

> > > ### Author Response · Authors · 2021-11-25
> > > **Response to Reviewer d8q8’s Feedback on Rebuttal**
> > >
> > > > + Q1: We see no comparison for LeViT that was available on arXiv in April, that is 6 month old enough (in my understanding mentioned policy doesn't exclude ArXiv). It is not crucial, however, could make the paper stronger.
> > >
> > > A1: We humbly clarify that the [policy](https://iclr.cc/Conferences/2022/ReviewerGuide) restricts the works, which are required to be compared with, as **“published (i.e., at a peer-reviewed venue)”**. Thus, LeViT[11], which was available on arXiv in April but published after July 22, 2021 (the date that ICCV’21 released the decision to authors), belongs to the “NOT required to compare with” case.
> > >
> > >
> > > > + Q2: The method of scaling up models is very expensive (retraining is required) and not novel. Therefore contribution is strictly experimental.
> > >
> > > A2:
> > >
> > > Regarding “The method of scaling up models is very expensive”, we humbly clarify that our adopted scaling method, iterative greedy search, is not more expensive as compared to other widely used scaling methods [1,2,3]. Specifically, [1,2] leverage grid search to scale up models, which will cost $m^n$ times of standalone training ($m$ is the number of options for each scaling factor and $n$ is the number of scaling factors), while our adopted iterative greedy search only cost $s \times n$ times of standalone training ($s$ is the number of steps in the iterative greedy search and $n$ is the number of scaling factors). Meanwhile, $s$ and $m$ are comparable ($s$ = 7 in our work and  $m$ = 5 or 7 in [2]), thus our adopted one’s scaling up cost is less expensive than or similar to theirs. Meanwhile, [3] also adopts an iterative greedy search approach to scale up models for video classification and classification, and a standalone training from scratch is also needed for each scaled model in it.
> > >
> > > Regarding “The method … not novel. Therefore contribution is strictly experimental”, we humbly remind that our work’s contributions, including Reviewer zFw4’s comments of “usefulness and practical friendliness” and “it is even a good learning experience for me” as well as Reviewer WMJX’s comment of “the analysis of the transferability of different scaling strategies on different hardware devices in this work is very positive and provides great inspiration to relevant fields”, match the ICLR criterion of  “bring value to the ICLR community when they convincingly demonstrate new, relevant, impactful knowledge” as mentioned in the ICLR Reviewer Guide. Finally, as we all recognize, there are quite many works (e.g., [4,5,6,7,8]) which share a similar goal of providing insights and usefulness to the community as our work, have been published in first-tier machine learning conferences like ICLR, and they have enhanced our understanding and inspired further innovations. It is worth noting that [4] received the best paper award in ICLR’19 and their empirical nature work’s “detailed and careful experimentation and analysis” were [especially appreciated by its reviewers and AC](https://openreview.net/forum?id=rJl-b3RcF7).
> > >
> > > > + Q3: The evidence of CNN scaling strategies is not addressed, as it is clear that it should be changed according to the parameters of the architecture.
> > >
> > > A3: We humbly remind that our claims about transferring the scaling strategies from CNNs to ViTs is “directly transferring the scaling strategies from CNNs to ViTs can lead to ambiguity and sub-optimal performance”, as illustrated in Section 3.2 of the latest manuscripts. Thus, “changed according to the parameters of the architecture” mentioned in your feedback is already not the “directly transferring” and it relies on human expertise and extra computation resources to tune the newly added scaling factors for ViTs, because the newly added ones do not belong to existing scaling factors for CNNs.
> > >
> > > > + Q4: Including CNN models into Fig 1 would be beneficial to see CNN models that are outperformed by the proposed scaling, being better than all CNN is not important.
> > >
> > > A4: Thank you for your suggestions! We have added “CNN models that are outperformed by the proposed scaling” in Figure 1 of this [anonymous link](https://ibb.co/Drprxrf) (we cannot upload the manuscript now). In particular, while currently the best ViT models are still inferior to state-of-the-art CNNs in terms of accuracy-FLOPs tradeoff, our scaled ViT leads to a comparable accuracy-FLOPs tradeoff than the famous efficient CNN called EfficientNet resulting from a compound scaling method.

---

> > > > ### Author Response · Authors · 2021-11-25
> > > > **Response to Reviewer d8q8’s Feedback on Rebuttal**
> > > >
> > > > > + Q5: Paper has useful experimental findings and agree with the reviewer zFw4 that paper is more on the technical report side and would be a great fit for a workshop.
> > > >
> > > > A5: Thank you for recognizing our useful experimental findings!  As we clarified in A2, we humbly remind that there are quite many works (e.g., [4,5,6,7,8]) which share a similar goal of providing insights and usefulness to the community as our work, have been published in first-tier machine learning conferences like ICLR, and they have enhanced our understanding and inspired further innovations. It is worth noting that [4] received the best paper award in ICLR’19 and “the detailed and careful experimentation and analysis” were [especially appreciated by its reviewers and AC](https://openreview.net/forum?id=rJl-b3RcF7).
> > > >
> > > >
> > > > We are happy to address any further comments or concerns, and wish you Happy Thanksgiving Holiday!
> > > >
> > > > *[1] Tan, Mingxing, and Quoc Le. "Efficientnet: Rethinking model scaling for convolutional neural networks." International Conference on Machine Learning. PMLR, 2019.*
> > > >
> > > > *[2] Bello, Irwan, et al. "Revisiting resnets: Improved training and scaling strategies." arXiv preprint arXiv:2103.07579 (2021).*
> > > >
> > > > *[3] Feichtenhofer, Christoph. "X3d: Expanding architectures for efficient video recognition." Proceedings of the IEEE/CVF Conference on Computer Vision and Pattern Recognition. 2020.*
> > > >
> > > > *[4] Frankle, Jonathan, and Michael Carbin. "The lottery ticket hypothesis: Finding sparse, trainable neural networks." arXiv preprint arXiv:1803.03635 (2018).*
> > > >
> > > > *[5] Neyshabur, Behnam, Hanie Sedghi, and Chiyuan Zhang. "What is being transferred in transfer learning?." Thirty-Fourth Conference on Neural Information Processing Systems. 2020.*
> > > >
> > > > *[6] Nguyen, Thao, Maithra Raghu, and Simon Kornblith. "Do wide and deep networks learn the same things? uncovering how neural network representations vary with width and depth." International Conference on Learning Representations, 2021.*
> > > >
> > > > *[7] Recht, Benjamin, et al. "Do imagenet classifiers generalize to imagenet?." International Conference on Machine Learning. PMLR, 2019.*
> > > >
> > > > *[8] Ying, Chris, et al. "Nas-bench-101: Towards reproducible neural architecture search." International Conference on Machine Learning. PMLR, 2019.*
> > > >
> > > > *[9] Touvron, Hugo, et al. "Training data-efficient image transformers & distillation through attention." International Conference on Machine Learning. PMLR, 2021.*
> > > >
> > > > *[10] Liu, Ze, et al. "Swin transformer: Hierarchical vision transformer using shifted windows." arXiv preprint arXiv:2103.14030 (2021).*
> > > >
> > > > *[11] Graham, Ben, et al. "LeViT: a Vision Transformer in ConvNet's Clothing for Faster Inference." arXiv preprint arXiv:2104.01136 (2021).*

---

### Official Review · Reviewer_zFw4 · 2021-11-07

**Correctness:** 3
**Technical Novelty And Significance:** 1
**Empirical Novelty And Significance:** 3
**Recommendation:** 3
**Confidence:** 3

**Main Review:**

+ The best part of the paper is about its "usefulness" and practical friendliness. The paper just studies the different factors that contributes to the model size/speed and accuracy (usually larger models on higher resolutions lead to better accuracy but slow down inference speed). An impressive part of the paper is that it shows measurements of speed beyond FLOPs -- it actually runs on various devices and measure the latency there. I think this practical mind is great.
+ It also shows good generalization of the finding. E.g. it can be transferred to more recent models (PiT), other tasks (object detection, video). This suggests solid improvement.
+ Overall the paper is well-written. The idea is clear, the organization is clean, and the figures are very illustrative. Overall it is even a good learning experience for me (e.g. knowing the latency break down of different devices) who is not so familiar with the hardwares.

- Biggest concern about this paper is novelty. It has some good findings, it shows good numbers, and it also teaches us something. However, the method (for search) is based on existing work, the application to other devices is quite straightforward (though surely requires a lot of hard work), and to me, it is more like a technical report about something is well executed, rather than an interesting "finding" that has a good deal of scientific merit.
- One more concern is about the word "scale". I don't think the study presented in this paper is about "scaling". To me the scaling is more about methods that scales *up* to larger models -- when people say about doing things "at scale", it indicates at "a large" scale. I come with the expectation that this paper can show how to scale our existing model to the next generation of model size, but only to find this paper is more about "speed-accuracy" trade-offs. So I would recommend replacing the term "scale" with "speed-accuracy trade-off" to be precise.

**Summary Of The Paper:**

The paper studies different hyper-parameters in the ViT design space (e.g., the number of layers, the dimensions per head, the number of head, the MLP ratio, the input image size). It starts with a very small, basic model, and then scales it up based on an iterative, greedy search. Each step it will select the best model (in terms of speed-accuracy trade-off) as the starting point of the next step. The "speed" aspect is firstly measured by FLOPs, and then later replaced by empirical measurements like latency on some devices. Overall, the resulting architectures show better speed-accuracy trade-offs than previous DEiT architectures, and shows consistent gains when transferred to other architectures (PiT) and tasks (e.g., object detection).

**Summary Of The Review:**

I think this paper is more on the "technical report" side, with quite limited novelty. I also find it gives me the wrong impression that the investigation is more about "scaling-up", but instead it just does search in the "small-to-base" scale regime. It definitely teaches me something and presents some interesting data points, but to me the scientific merit is not enough to warrant a publication.

---

> ### Author Response · Authors · 2021-11-23
> **Response to Reviewer zFw4**
>
> > + Q1: Biggest concern about this paper is novelty. It has some good findings, it shows good numbers, and it also teaches us something. However, the method (for search) is based on existing work, the application to other devices is quite straightforward (though surely requires a lot of hard work), and to me, it is more like a technical report about something is well executed, rather than an interesting "finding" that has a good deal of scientific merit.
>
> A1: Thank you for appreciating our findings inside this work and the corresponding “usefulness and practical friendness”!  We humbly clarify that we did not claim the novelty of the method (for search), and instead our main contribution is the study and analysis of the scaled architectures and their corresponding transferability across different devices, ViT variants, and recognition tasks (i.e., the “usefulness and practical friendness” mentioned by you).  For example, we show that a scaled vanilla ViT model without bells and whistles can achieve comparable or superior accuracy-efficiency trade-off than most of the latest ViT variants, which can provide a new perspective for designing ViTs with more favorable accuracy-efficiency trade-offs.
>
> Furthermore, we hope to humbly remind that our work’s contributions, including your mentioned “usefulness and practical friendliness” and “it is even a good learning experience for me” as well as Reviewer WMJX’s comment of “the analysis of the transferability of different scaling strategies on different hardware devices in this work is very positive and provides great inspiration to relevant fields”, match the ICLR criterion of  “bring value to the ICLR community when they convincingly demonstrate new, relevant, impactful knowledge” mentioned in the ICLR Reviewer Guide. Finally, as we all recognize, there are quite many works (e.g., [5,6,7,8]) which share a similar goal of providing insights and usefulness to the community as our work, have been published in first-tier machine learning conferences like ICLR, and they have enhanced our understanding and inspired further innovations.
>
> > + Q2: One more concern is about the word "scale". I don't think the study presented in this paper is about "scaling". To me the scaling is more about methods that scales up to larger models -- when people say about doing things "at scale", it indicates at "a large" scale. I come with the expectation that this paper can show how to scale our existing model to the next generation of model size, but only to find this paper is more about "speed-accuracy" trade-offs. So I would recommend replacing the term "scale" with "speed-accuracy trade-off" to be precise.
>
> A2: Thank you for your comment! While your mentioned “how to scale our existing model to the next generation of model size” is an important problem, we would like to humbly point out that scaling the models **down** is equally useful for deploying them into resource-constrained daily-life devices [1,2,3,4], which is one of our exploration goals, e.g., “can these scaling strategies be transferred across different real hardware devices?”. As such, in our humble opinion, the “scaling” terminology still holds for our work as we consider both scaling up and down ViTs. We will better clarify in our final version.
>
> *[1] Lin, Ji, et al. "Memory-efficient Patch-based Inference for Tiny Deep Learning." Thirty-Fifth Conference on Neural Information Processing Systems, 2021.*
>
> *[2] David, Robert, et al. "Tensorflow lite micro: Embedded machine learning on tinyml systems." arXiv preprint arXiv:2010.08678 (2020).*
>
> *[3] Banbury, Colby, et al. "Micronets: Neural network architectures for deploying tinyml applications on commodity microcontrollers." Proceedings of Machine Learning and Systems 3 (2021).*
>
> *[4] Coiffier, Guillaume, Ghouthi Boukli Hacene, and Vincent Gripon. "ThriftyNets: Convolutional Neural Networks with Tiny Parameter Budget." IoT 2.2 (2021): 222-235.*
>
> *[5] Neyshabur, Behnam, Hanie Sedghi, and Chiyuan Zhang. "What is being transferred in transfer learning?." Thirty-Fourth Conference on Neural Information Processing Systems. 2020.*
>
> *[6] Nguyen, Thao, Maithra Raghu, and Simon Kornblith. "Do wide and deep networks learn the same things? uncovering how neural network representations vary with width and depth." International Conference on Learning Representations, 2021.*
>
> *[7] Recht, Benjamin, et al. "Do imagenet classifiers generalize to imagenet?." International Conference on Machine Learning. PMLR, 2019.*
>
> *[8] Ying, Chris, et al. "Nas-bench-101: Towards reproducible neural architecture search." International Conference on Machine Learning. PMLR, 2019.*

---

### Decision · Program_Chairs · 2022-01-20

**Decision:**

Reject

**Comment:**

This paper explores strategies for scaling vision transformers that can be transferable across hardware devices and ViT variants. While it presents some interesting observations as well as a useful practical guide, multiple reviewers expressed major concerns over the novelty and significance of the methods and findings. Besides novelty and significance, there are also some concerns about comparison with existing work as well as clarity of the presentation.